# Biological Response of Irisin Induced by Different Types of Exercise in Obese Subjects: A Non-Inferiority Controlled Randomized Study

**DOI:** 10.3390/biology11030392

**Published:** 2022-03-02

**Authors:** Andrea D’Amuri, Valeria Raparelli, Juana Maria Sanz, Eleonora Capatti, Francesca Di Vece, Filippo Vaccari, Stefano Lazzer, Giovanni Zuliani, Edoardo Dalla Nora, Luca Maria Neri, Angelina Passaro

**Affiliations:** 1Department of Translational Medicine, University of Ferrara, Via Luigi Borsari 46, 44121 Ferrara, Italy; dmrndr@unife.it (A.D.); valeria.raparelli@unife.it (V.R.); giovanni.zuliani@unife.it (G.Z.); edoardo.dallanora@unife.it (E.D.N.); 2Medical Department, University Hospital of Ferrara Arcispedale Sant’Anna, Via A. Moro 8, 44124 Ferrara, Italy; cptlnr@unife.it (E.C.); f.divece@ospfe.it (F.D.V.); 3Department of Chemical, Pharmaceutical and Agricultural Sciences, University of Ferrara, Via Luigi Borsari 46, 44121 Ferrara, Italy; juana.sanz@unife.it; 4Department of Medical Sciences, School of Sport Science, University of Udine, Piazzale M. Kolbe 4, 33100 Udine, Italy; filippo.vaccari@uniud.it (F.V.); stefano.lazzer@uniud.it (S.L.); 5Laboratory for Technologies of Advanced Therapies (LTTA)—Electron Microscopy Center, University of Ferrara, Via Luigi Borsari 46, 44121 Ferrara, Italy; 6Research and Innovation Section, University Hospital of Ferrara Arcispedale Sant’Anna, Via A. Moro 8, 44124 Ferrara, Italy

**Keywords:** irisin, obesity, interval training, endurance training, sex

## Abstract

**Simple Summary:**

Among healthy male and female obese individuals undergoing a 12-week aerobic exercise program with either moderate intensity endurance or high-intensity interval training for losing weight, a reduction of circulating irisin was observed. Irisin is an important adipo-myokine implicated in the regulation of energy metabolism and cardiovascular health. Sex differences in the circulating levels of this biomarker have been previously reported and are likely related to the different anthropometric features between the sexes. A sex-specific modulation of circulating irisin levels should be further explored to tailor sex-specific training approaches for improving the cardiovascular health of obese subjects.

**Abstract:**

**Background:** Weight loss through physical exercise is warranted among obese individuals. Recently, a greater benefit in cardiorespiratory fitness was achievable with high-intensity interval training (HIIT) as compared with moderate intensity continuous training. The beneficial effect of training on CV health might be related to a specific modulation of circulating irisin, an adypo-myokine implicated in the regulation of energy expenditure. **Methods****:** The present study investigates the circulating plasma levels of irisin at baseline and in response to 12-week of training program either with HIIT or moderate-intensity continuous training (MICT) among young female and male obese subjects. Clinical, anthropometric, and training characteristics for each participant were available. A sex-disaggregated data for circulating plasma levels of irisin pre- and post-training are provided as well as an adjusted multivariate linear regression model to identify the determinants of post-training irisin levels. **Results:** Data from a total of 32 obese healthy individuals (47% female, mean age 38.7 years, mean BMI 35.6 kg/m^2^), randomized in a 1:1 manner to HIIT or MICT were analyzed. Circulating plasma levels of irisin similarly and significantly decreased in both MICT and HIIT interventional groups. Females had higher post-exercise irisin levels than males (6.32 [5.51–6.75] vs. 4.97 [4.57–5.72] μg/mL, *p* = 0.001). When stratified by an interventional group, a statistically significant difference was observed only for the MICT group (male, 4.76 [4.20–5.45] μg/mL vs. female 6.48 [4.88–6.84] μg/mL *p* = 0.03). The circulating post-training level of irisin was independently associated with post-training fat-free mass (β −0.34, 95% confidence interval, CI −0.062, −0.006, *p* = 0.019) in a model adjusted confounders. When female sex was added into the adjusted model, it was retained as the only factor independently associated with irisin levels (β 1.22, 95% CI, 0.50, 1.93, *p* = 0.002). **Conclusions:** In obese healthy subjects, circulating irisin levels were reduced in response to 12-weeks of exercise involving either HIIT or MICT. A sex-specific differences in circulating irisin levels at baseline and as biological response to chronic exercise was described. Sex-specific biological response of irisin to exercise should be further explored to tailor sex-specific training approaches for improving the cardiovascular health of obese healthy subjects.

## 1. Introduction

The burden of obesity is a matter of immediate concern worldwide with high healthcare-related costs due predominantly to the increase in type II diabetes, cardiovascular diseases, and all-cause mortality [1,2,3]. Currently, exercise is recommended as a first line treatment of obesity [4] since it induces weight loss, fat mass reduction, improvement of major metabolic and cardiovascular risk [5] factors, and increases cardiorespiratory fitness [6]. The current guidelines suggest that obese people perform 250–400 min per week of moderate intensity continuous training (MICT) [4], as well as with an intensity ranging from 46 to 63% of maximal oxygen uptake (VO2max) [7]. Nevertheless, we have recently demonstrated that high-intensity interval training (HIIT), an activity characterized by short bouts of high-intensity exercise (up to 100% of VO2max) alternating with periods of low-intensity exercise has emerged as an alternative non-inferior option for obesity treatment with the additive advantages of increasing cardiorespiratory fitness (CRF), an important marker of cardiovascular health in younger individuals, improving autonomic function [8], and requiring less time to devote [9].

The mechanisms underlying the physical benefits of exercise in obesity management are still only partially understood. During muscle contraction, myocytes act as a secretory organ and release hormones called myokines that work in an autocrine, paracrine, and endocrine manner. Irisin is a short-life myokine cleaved from membrane fibronectin type III domain-containing protein 5 (FNDC5) and regulated by peroxisome proliferator-activated receptor (PPAR)-γ coactivator-1α (PGC-1α) [10], which likely mediates some of the exercise-related health benefits. Thus, preclinical studies showed that exercise-induced release of irisin promotes the browning of adipose tissue and improves glucose and lipid metabolism. Experimental data reported that prolonged HIIT was more efficient than MICT to induce irisin secretion [11] in lean rats, to suppress adipogenesis and inducing browning of adipose tissue in obese rats [12], and to improve glycemic control and lipid profile in diabetic rats [13]. Whether the muscles’ release of irisin and its beneficial effect in humans varies depending on the type of exercise (i.e., acute bout exercise versus long term training) is currently a matter of debate. While Bostrom et al. originally described irisin release in healthy males after 10-weeks of MICT [14], others disconfirmed this association. On the other hand, prolonged training resulted in an only modest or no increase of plasma irisin [15,16,17,18]. Moreover, prior experimental work suggested that the exercise-induced release of irisin varies over time: the single-time acute exercise increases plasma levels of irisin [10,19,20,21,22], but when the acute bouts are repeated constantly over several weeks, the levels flatten and are no longer detectable [10,20].

Finally, data on the effect of HIIT and MICT in modulating the plasma levels of irisin are scanty and controversial, with a study reporting a reduced irisin level after 12 weeks of HIIT in overweight pre-diabetic young males [23] and another work showing an increase of plasma irisin after four weeks of HIIT in overweight/obese diabetic older people of both sexes [24]. To our knowledge, no study has ever compared the biological response of circulating levels of irisin production between HIIT and MICT in obese male and female young healthy subjects.

To fill in this gap, we assessed the plasma levels of irisin at baseline and after an aerobic exercise program comparing HIIT versus MICT and describing the specific determinants of irisin level among obese young healthy individuals.

## 2. Materials and Methods

The data underlying this article will be shared upon reasonable request to the corresponding author.

### 2.1. Study Design and Population

For this secondary analysis aimed at studying the biological response of irisin to long-term aerobic exercise, we used data that come from a randomized, single-blind, single-centre, parallel-group, non-inferiority trial previously published that compares the efficacy of 12-weeks of HIIT and MICT in weight loss among obese (i.e., BMI between 30 and 55 kg/m^2^) healthy adults (≤50 years). The study design and the main findings concerning the primary objective of the original study has been already published [25]. Briefly, the study was performed at the Exercise Physiology Laboratory of the University of Udine, Italy, between September 2017 and June 2018. All subjects followed a tailored hypocaloric diet. Changes in anthropometrics, arterial blood pressure, metabolic parameters (i.e., lipid and glucose measurements), and cardiorespiratory fitness (CRF) have been assessed. The study was approved by the Institutional Review Board of the Friuli-Venezia-Giulia Region. All patients provided their written informed consent to participate. Thirty-two subjects were recruited and randomized to either the HIIT or MICT group. Individuals followed a 12-week program of aerobic physical exercise (three sessions per week). Training sessions were supervised by a research assistant to guarantee adherence to at least 90% of the exercise sessions. Training regimes to maximize the improvements of oxygen uptake max (VO2max) were selected as previously performed by Buchheit et al. [26] (i.e., the intensity was close to 100% of the velocity corresponding to oxygen peak uptake (VO2peak) and to around 50% during recovery intervals. Finally, a 3-min intervals protocol with a work/recovery ratio ≤ 1 was used. HIIT and MICT programs per session had 10 min of warm-up (50% VO2peak) followed by 5 min of cool-down (50% VO2peak). The duration of the training sessions was scheduled for achieving equal amounts of energy expenditure regardless of the interventional group assigned [20 kJ (4.78 Kcal) per FFM kg]. Every month, physical capacities were assessed, and the physical training intensity was individually adjusted. Data for the number of training sessions, duration of any session, and energy expenditure were obtained.

For the present analysis the following characteristics were used: age, sex, body mass index (BMI), heart rate, systolic and diastolic blood pressure, daily calorie intake, training program features, CRF (assessed as VO2 peak and max, also scaled for body weight), body composition (defined by fat mass (FM) and fat-free mass (FFM) expressed as both kilograms and percentage using a two-compartment model). Furthermore, laboratory measures such homeostasis model assessment (HOMA-IR), lipid parameters (i.e., total cholesterol, triglycerides, high-density lipoprotein cholesterol, low-density lipoprotein cholesterol), C reactive protein, hormones (i.e., cortisol, insulin-like growth factor 1), creatinine levels, and Irisin were also included. All measurements and samples were obtained at baseline data collection (BDC) and post-training data collection (PTDC).

### 2.2. Biological Samples

After overnight fasting, blood samples were collected from each subject and centrifuged in the absence or presence of EDTA to obtain serum and plasma. Aliquots were stored at −80 °C. Plasma irisin was measured by ELISA (Adipogen). The coefficient of variation was <7% and <10% for the intra-assays.

Finally, FNDC5 gene expression in muscle tissue was available for all participants. Muscle tissues were obtained from the vastus lateralis muscle after anesthesia of the skin using lidocaine (2%). A small incision was made to penetrate skin and fascia, and then a biopsy sample was harvested with a microneedle (Tru-cut Histocore, 12 G, Biomed Instrument and product GmbH, Türkenfeld, Germany). A fragment of the sample was quickly immerged in lysis solution (Purezool, Bio Rad, Milan, Italy) and stored at −80 °C. Tissue sample was disrupted and homogenized using a tissue ruptor (Qiagen, Milan, Italy). Successively, RNA was isolated using Aurum Total RNA Mini kit (Bio Rad) and stored at −80 °C until use. cDNA was prepared from RNA using a High-Capacity cDNA Reverse Transcription Kit (Life Technologies Italia, Italy). Gene expression was measured by Real-Time PCR (StepOnePlus™, Applied Biosystems, Life Technologies Italia) using pre-designed TaqMan gene expression assays (Applied Biosystems, Life Technologies Italia): FNDC5, Hs00401 006 m1; Ribosomal Protein S13 (RPS13), Hs01 011 487 g1. The amplification reaction was performed in duplicate in 48-well plates. Data were expressed as fold increase of the target cDNA in the samples after normalized by the 2(ΔCt) method using RPS13 as housekeeping gene and a reference sample.

### 2.3. Statistical Analysis

All continuous variables were tested for normality with the Shapiro-Wilk test. Continuous variables with normal distribution were reported as mean ± standard deviation (SD), non-parametric variables as median and interquartile range (IQR). Between-group comparisons were performed using T test for normally-distributed variables and using an appropriate non-parametric test for non-normally distributed variables (Mann-Whitney U test or Kruskal–Wallis H test). Categorical variables were reported as count and percentages. Between-group comparisons were made using a χ^2^ test, or a Fisher’s exact test if any expected cell count was <5. When appropriated, the Wilcoxon Signed Rank U test was used to compare two groups with dependence but non-normal distribution. We used the Spearman rank correlation test to explore the bivariate association between irisin levels and other parameters of interests. Based on biological plausibility, a set of covariates were included in a linear regression multivariate stepwise analysis to identify the factors independently associated with post training irisin level. Furthermore, a sex-stratified analysis was also conducted. A two-sided *p*-value < 0.05 was considered statistically significant. All analyses were performed using SPSS v. 25.0 statistical software (International Business Machines, New York, NY, USA).

## 3. Results

### 3.1. Clinical, Physical and Antropometric Features of the Study Population

A total of 32 obese healthy individuals (47% female, mean age 38.7 years, mean BMI 35.6 kg/m^2^). The main characteristics of the study population by an interventional group are summarized in Table 1.

As expected, due to randomization, baseline characteristics of the 32 patients randomized to HIIT or MICT were similar between interventional groups. The overall effect of the training program (regardless of type) are summarized in Appendix A. Regardless of training type, beneficial changes of anthropometric parameters and body composition (reduction of fatty mass, lipid profile (i.e., reduction of LDL cholesterol) were achieved among the overall cohort (Table 1 and Appendix A).

### 3.2. Irisin Levels in HIIT and MICT

Circulating irisin levels are different after 12 weeks of training compared to baseline (BDC, 6.46 [5.59–7.02] μg/mL vs. PTDC, 5.62 [4.78–6.32], *p* = 0.0001). Both baseline (MICT, 5.89 [5.23–6.77] μg/mL vs. HIIT, 6.47 [5.53–8.24] μg/mL) and post-training (MICT, 5.18 [4.51–6.45] μg/mL vs. HIIT, 6.01 [4.93–6.16] μg/mL) levels of irisin did not differ between interventional groups.

Circulating plasma levels of irisin similarly and significantly decreased in both MICT and HIIT (Figure 1) interventional groups.

Regardless of interventional group, female obese individuals had higher baseline irisin levels than male counterpart (6.75 [6.07–8.24] vs. 5.58 [4.92–6.29] μg/mL, *p* = 0.005) (Figure 2a). Sex differences were also found for post-training irisin levels: females had higher post-exercise irisin levels than males (6.32 [5.51–6.75] vs. 4.97 [4.57–5.72] μg/mL, *p* = 0.001) (Figure 2a).

When data were stratified by interventional group, a statistically significant sex difference in baseline irisin was observed for HIIT group (female 7.16 [6.44–8.36] μg/mL vs. male, 6.11 [5.58–4.76] μg/mL, *p* = 0.003) while post-training irisin was lower in males than females only in MICT group (female 6.48 [4.88–6.84] μg/mL vs. male, 4.76 [4.20–5.45] μg/mL, *p* = 0.03) (Figure 2b).

Sex-specific changes of irisin from baseline to 12 weeks are depicted in Figure 3. Here, a not statistically significant greater reduction among females underwent HIIT was observed.

### 3.3. Factors Associated with Plasma Levels of Irisin

Bivariate correlation analyses (Appendix A) were performed to explore the association between irisin levels and the multidimensional features available for all participants. As depicted in Appendix A, clinical characteristics and demographics were not correlated with either baseline or post-training irisin levels except for biological sex. Notably, when considering physical capacities and training measures, the CFR expressed by VO2 peak (mL) was inversely correlated with post-trainig irisin regadless of the specific traing program, suggesting that the beneficial effect in terms of reduction of irisin is associated with the improvement of CFR (Appendix A). The main correlations were found between body composition and post-training irisin with free fat mass that significantly and inversily correleted with irisin (Appendix A). No significant correlations with metabolic measures and irisin were observed (Appendix A). The gene expression of FDNC5 of the muscle did not differ between interventional groups [FNDC5 gene expression (relative quantification, RQ): 1.3 ± 0.4 vs. 1.3 ± 0.6, *p* = 0.59]. Notably, similar correlation matrixes were obtained when we tested the bivariate association between FDNC5 gene expression and all available patient’s measures. Finally, significant changes in antropometrics, body composition and metabolic measures were observed when grouping the cohort based on the median change of irisin achieved (Appendix A).

Therefore, to better identify the factors independently associated with post-training levels of irisin, we performed a multivariable linear regression analysis. The circulating post-training level of irisin was independently associated with post-training FFM (kg) (β −0.34, 95%CI −0.062, −0.006, *p* = 0.019) in a model adjusted for age, HOMA, creatinine, training type, weight, FM (kg), VO2 peak/weight. When female sex was added into the adjusted model, it was retained as the only factor independently associated with irisin level (β 1.22, 95%CI, 0.50, 1.93, *p* = 0.002).

## 4. Discussion

In this study, a 12-week HIIT and MICT in healthy obese individuals induced a similar decrease of irisin plasma levels suggesting an adaptative long-term biological response to exercise. Sex-specific differences in pre- and post-training levels of irisin were evident between HIIT and MICT, with a trend to a greater decrease of irisin in males after MICT. The only factor independently associated with post training low irisin level was the increased FFM after training.

Several preclinical and clinical studies attempted to understand the relationship between irisin and obesity as well as the kinetics of adipo-myokine [27]. Animal studies demonstrating important metabolic benefits have been convincing [14], but whether these findings translate to humans remains unclear. Prior studies in healthy non-smoking adults reported an acute increase of irisin levels in response to exercise with a greater increase after maximal workload, suggesting that irisin circulating level represents a function of muscle energy demand [28]. Despite irisin being considered a marker of energy expenditure, obese individuals had paradoxically increased circulating plasma levels as compared to normal weight individuals. In fact, a reduced irisin levels were reported as compared to morbidly obese individuals. While increased fat deposits results in elevated irisin levels, it seems that irisin itself loses the ability to continue to exert its effects in a proper fashion. Similarly with other conditions such as insulin resistance or leptin resistance, obese individuals might develop irisin resistance. Prior work conducted among only male obese pre-diabetic individuals, older than the subjects enrolled in the present study, reported a similar post-exercise decrease in irisin level [21]. In light of this, the decrease of irisin levels observed in our study after a long aerobic training program can be the result of the weight loss and restructure of body composition obtaining through HIIT and MICT. Thus, we cannot exclude that the beneficial effect of 12-week aerobic training on cardiovascular health might be due to enhanced sensitivity of irisin action and/or a reduced need for irisin secretion after long-term training. Finally, our findings are in accordance with a metanalysis [29] reporting how chronic exercise training was associated with decreased circulating irisin levels in healthy untrained adults (predominantly when looking at RCTs). This study extends the prior literature providing a comprehensive analysis that assessed the biological response of irisin in relation to multiple relevant factors (i.e., clinical, anthropometric, and metabolic/hormonal measures) as well as the gene expression of FDNC5 to better inform our understanding of in-vivo variations of this crucial adypo-myokine implicated in regulation of energy expenditure. Further mechanistic studies are warranted to clarify the balance between irisin, FDNC5 expression, and chronic aerobic exercise to implement future intervention strategy for managing the health of young obese individuals.

The importance of integrating sex as a biological variable in clinical studies has been constantly remarked by international scientific association as it boosts knowledge translation and results in high-quality and equitable science [30]. A sexual dimorphism has been reported for irisin both at rest and in response to acute exercise [31,32]. The differences in anthropometric and physiological parameters such as the distribution of fat (white and brown) or sex hormones among male and female young adults might explain the sex differences in circulating levels of irisin. Interestingly, we observed some sex-specific differences in circulating levels of irisin at baseline and after specific chronic anaerobic training programs that have not previously reported. Also of note is the fact that the effect of anthropometric parameters on irisin levels in our cohort disappear when the multivariable model was adjusted for sex, suggesting that all of the features of being a female subjects can explain the variation in irisin levels.

The present findings should be interpreted in the light of several limitation. The small sample size and the monocentric study design limit the generalizability of our findings. Although sex differences were observed, the study was designed to detect sex differences in circulating plasma levels of irisin. Notably, currently available ELISA assays for quantifying circulating irisin levels still lack quality and accuracy, and the measured values are very dissimilar between the different assays [33]. This may partially explain the discrepant data found by different studies. As we only tested the muscle gene expression of FDNC5, we cannot disentangle whether the reduction in circulating levels of irisin may be related to variation in the adipose tissue gene expression of FDNC5. Finally, the beneficial effect of exercise on irisin reduction might be also partially driven by the hypocaloric diet followed by the participants as per protocol. However, the relationship between energy restriction, weight loss, and circulating irisin has been previously explored with conflicting results [34,35,36,37]. Of note, we have previously reported how the energy intake was similar between the intervention groups, therefore suggesting that the diet effect should be attenuated in this context.

## 5. Conclusions

Long-term aerobic training either HIIT or MICT are equally effective in reducing weight loss and circulating level of irisin. A sex-specific modulation of circulating irisin levels should be further explored to tailor sex-specific training approaches for improving the cardiovascular health of obese healthy subjects.

## Figures and Tables

**Figure 1 biology-11-00392-f001:**
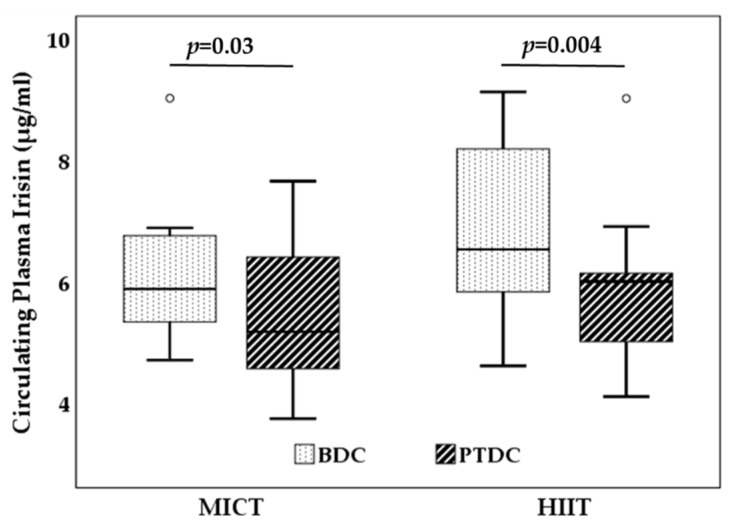
Changes in circulating level of irisin after the aerobic training of 12 weeks: MICT and HIIT groups. BDC, baseline data collection; PTDC, post training data collection; MICT, moderate-intensity continuous training; HIIT, high-intensity interval training.

**Figure 2 biology-11-00392-f002:**
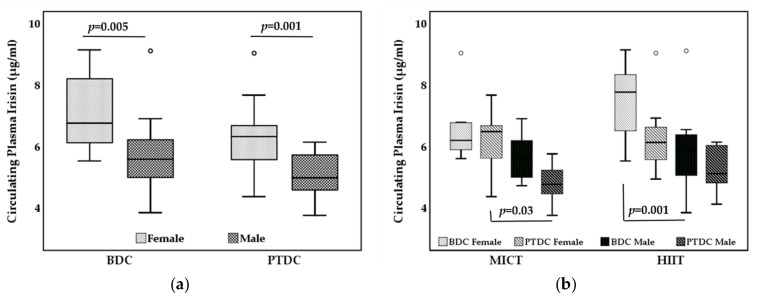
Sex-stratified circulating plasma irisin (**a**) before and after 12 weeks of training and (**b**) and stratified by interventional group. BDC, baseline data collection; PTDC, post training data collection; MICT, moderate-intensity continuous training; HIIT, high-intensity interval training.

**Figure 3 biology-11-00392-f003:**
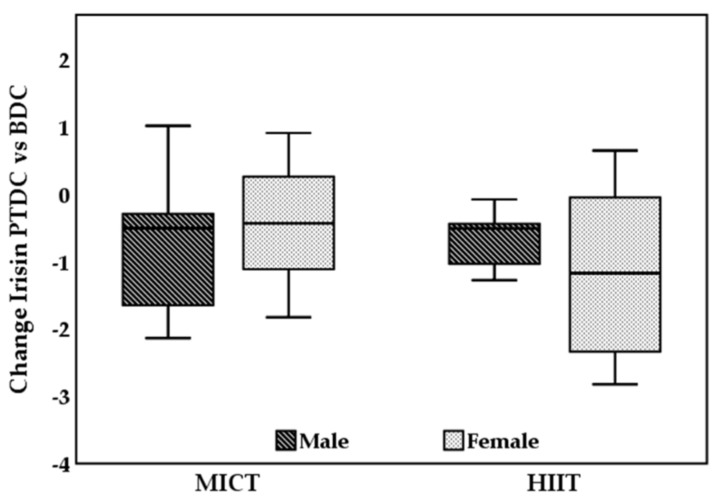
Sex-specific changes in circulating level of irisin after the aerobic training of 12 weeks with MICT group or HIIT. BDC, baseline data collection; PTDC, post training data collection; MICT, moderate-intensity continuous training; HIIT, high-intensity interval training.

**Table 1 biology-11-00392-t001:** Clinical, anthropometric and physical features by interventional group ^1^.

**Variables**	**MICT Group (*n* = 16)**		**HIIT Group (*n* = 16)**	
Age, years	37.2 ± 9.1		40.1 ± 7.0	
Female sex, *n* (%)	7 (43.8)		8 (50.0)	
Prior or current smokers, *n* (%)	2 (12.5)		3 (18.8)	
Prior weight loss attempts, *n* (%)	11 (68.8)		11 (68.8)	
Family history of diabetes, *n* (%)	3 (18.8)		4 (25.0)	
	**BDC**	**PTDC**	**BDC**	**PTDC**
Weight (kg)	105.7 ± 17.4	98.5 ± 17.9	103.4 ± 10.9	97.6 ± 10.1
BMI (kg/m^2^)	35.7 [32.3–35.4]	33.1 [30.7–35.4]	34.3 [32.5–37.6]	32.0 [29.8–36.3]
VO2peak (mL)	2966.5 [2438.0–3520.5]	3263.5 [2496.5–3571.5]	2724.5 [2305.0–3541.0]	3302.5 [2566.0–3992.0]
FM (Kg)	37.7 ± 10.9	32.4 ± 9.1	38.4 ± 8.2	32.9 ± 10.0
FM (%)	35.8 [28.2–42.7]	31.8 [24.5–40.9]	39.1 [31.4–43.1]	32.6 [26.6–42.2]
FFM (Kg)	69.4 ± 15.5	68.6 ± 16.3	65.1 ± 11.7	64.7 ± 11.0
FFM (%)	64.2 [57.2–71.8]	68.2 [59.1–75.3]	60.9 [57.0–69.0]	67.4 [58.4–73.4]
Creatinine (mg/dL)	0.7 ± 0.1	0.8 ± 0.1	0.8 ± 0.1	0.8 ± 0.1
HOMA Index	1.8 [1.4–3.1]	1.4 [1.1–2.1]	2.0 [1.6–3.0]	1.4 [1.2–2.8]

^1^ MICT, moderate-intensity continuous training; HIIT, high-intensity interval training; BMI, body mass index; FFM, fat-free mass; FM, fat mass; VO2peak, peak oxygen uptake; BDC, baseline data collection; PTDC, post training data collection. Data are expressed as media ± standard deviation or median [interquartile range] unless otherwise specified.

## Data Availability

Data are available through the corresponding author upon request and for justified reasons.

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
