# Peer review of "Biological Response of Irisin Induced by Different Types of Exercise in Obese Subjects: A Non-Inferiority Controlled Randomized Study"

_biology, 2022, doi:10.3390/biology11030392_

Round 1
Reviewer 1 Report
This work by D'Amuri et al. intends to compare the circulating levels of the myokine Irisin in response to 12-weeks of different types of exercise (HIIT vs MICT) in a group of obese individuals.
Their results show that while the subjects lost weight and showed an overall improvement of metabolic and other health markers after the 12-week period, the levels of Irisin in their blood were reduced, with no changes in FNDC5 gene expression in their muscle tissue (a small sample was taken at the end of the exercise regimen).
While these results are straight forward, I find several issues that need to be addressed.
- These individuals were also given a hypocaloric diet throughout the exercise protocol, which resulted in significant fat mass loss. The reduction in plasma irisin could be simply a reflection of the reduction in fat mass. I wonder if the lower caloric intake could be a confounding factor in the experimental design as all individuals were subjected to it and what could be happening is a synergistic effect with exercise. Is there anyway you can add data from a group not given a different diet? Please address this in the discussion.
- I am also confused by the data in figure 3 that shows no differences in Irisin before and after exercise, when the data are compared according to sex. Is this not the same data that shows reduced Irisin in plasma in figures 2? I understand that these are raw values compared to a subtraction (fig 2 vs 3) but even considering this technical issue, it is expected that the difference should be significant when you subtract the values post-exercise from pre. Please clarify this issue.
- What is the reason for the blood samples to be collected after the last bout of exercise? (according to methods).
- There is no mention of the FNDC5 gene expression results in the main text, they are just in supplementary figures/tables. I think this should be addressed as the circulating levels of irisin are reduced, then where is that reduction coming from if not from the muscle? Should be discussed
Author Response
- These individuals were also given a hypocaloric diet throughout the exercise protocol, which resulted in significant fat mass loss. The reduction in plasma irisin could be simply a reflection of the reduction in fat mass. I wonder if the lower caloric intake could be a confounding factor in the experimental design as all individuals were subjected to it and what could be happening is a synergistic effect with exercise. Is there anyway, you can add data from a group not given a different diet? Please address this in the discussion.
REPLY: We thank the reviewer for the comment. We agree that it is plausible that hypocaloric diet could have had a synergistic effect on the circulating level of irisin. Nevertheless, the relationship between energy restriction, weight loss and circulating irisin has been previously explored with conflicting results [see references now added in the main manuscript].
Notably, in data already published of the original interventional study, we observed a similar weight loss achieved with no differences in the energy intake between the interventional groups, suggesting that, even if plausible, the independent effect of hypocaloric diet should be attenuated.
We have now included a comment in the limitation section of the study.
…[Finally, the beneficial effect of exercise on irisin reduction might be also partially driven by the hypocaloric diet followed by the participants as per protocol. However, the relationship between energy restriction, weight loss and circulating irisin has been previously explored with conflicting results [34-37]. Of note, we have previously reported how the energy intake was similar between the intervention groups therefore suggesting that the diet effect should be attenuated in this context]...
Added References:
- Park K.H,; Zaichenko L.; Brinkoetter M.; Thakkar B.; Sahin-Efe A.; Joung K.E.; Tsoukas M.A.; Geladari E.V.; Huh J.Y.; Dincer F.; Davis C.R.; Crowell J.A.; Mantzoros C.S. Circulating irisin in relation to insulin resistance and the metabolic syndrome. J Clin Endocrinol Metab 2013, 98, 4899-907.
- Crujeiras A.B.; Pardo M.; Arturo R.R.; Navas-Carretero S.; Zulet M.A.; Martínez J.A.: Casanueva F.F. Longitudinal variation of circulating irisin after an energy restriction-induced weight loss and following weight regain in obese men and women. Am J Hum Biol 2014, 26, 198-207.
- Fukushima Y.; Kurose S.; Shinno H.; Thi Thu H.C.; Takao N.; Tsutsumi H.; Hasegawa T.; Nakajima T.; Kimura Y. Effects of Body Weight Reduction on Serum Irisin and Metabolic Parameters in Obese Subjects. Diabetes Metab J 2016, 40, 386-395.
- Kim H.J.; Lee H.J.; So B.; Son J.S.; Yoon D.; Song W. Effect of aerobic training and resistance training on circulating irisin level and their association with change of body composition in overweight/obese adults: a pilot study. Physiol Res 2016, 65, 271-9.
2. I am also confused by the data in figure 3 that shows no differences in Irisin before and after exercise, when the data are compared according to sex. Is this not the same data that shows reduced Irisin in plasma in figures 2? I understand that these are raw values compared to a subtraction (fig 2 vs 3) but even considering this technical issue, it is expected that the difference should be significant when you subtract the values post-exercise from pre. Please clarify this issue.
REPLY: We thank the reviewer for this comment. As the reviewer pointed out while the figure 2 considered only the raw data at different timepoints (baseline vs after 12-week intervention), the figure 3 depicts the relative variation of irisin [post-training – baseline]. We have proposed the data using these 2 approaches to give a more feasible representation of the change in irisin by sex and by type of intervention. Although not significant, the change is consistent with the absolute numbers that are proposed in Figure 2.
3. What is the reason for the blood samples to be collected after the last bout of exercise? (according to methods).
REPLY: We thank the reviewer for the comment. As the aim of this secondary analysis was to understand the effect of long-term aerobic exercise on irisin levels among obese young adults, we decided to measure the biological response of the biomarker at the end of the complete training program (i.e., after 12 weeks). In fact, as stated in the introduction, there is a current debate on the differential effect of single bout of exercise versus long-term training with paucity of data on the latter.
4. There is no mention of the FNDC5 gene expression results in the main text, they are just in supplementary figures/tables. I think this should be addressed as the circulating levels of irisin are reduced, then where is that reduction coming from if not from the muscle? Should be discussed
REPLY: As suggested, we have implemented the results on the FNDC5 gene expression of the muscle tissue in the main text. It would be interesting to understand if the reduction of irisin results from a variation in the adipose tissue expression of FNDC5. Unfortunately, we could not verify the adipose tissue gene expression of FNDC5 among the participants, due to technical issues in the extraction procedure that made the assessment unreliable. We have discussed this aspect as a limitation of the study.
…[The gene expression of FDNC5 of the muscle did not differ between interventional groups [FNDC5 gene expression (relative quantification, RQ): 1.3±0.4 vs 1.3±0.6, p=.59]. Notably, similar correlation matrixes were obtained when was tested the bivariate association between FDNC5 gene expression and all available patient’s measures]...
… [As we only tested the muscle gene expression of FDNC5 we cannot disentangle whether the reduction in circulating levels of irisin may be related to variation in the adipose tissue gene expression of FDNC5].
Reviewer 2 Report
Biological response of irisin induced by different types of exercise in obese subjects: a non-inferiority controlled randomized study
D’Amuri A et al aimed to “assess the plasma levels of irisin at baseline and after aerobic exercise program comparing HIIT versus MICT, as well as described the specific determinants of irisin level among obese young healthy individuals.”
The paper is interesting and well written and helps to disentangle the role of Irisin in biological systems. However, there are a few issues that need attention.
Major issues
- A clear objective should be stated. A non-inferiority controlled randomized study is mentioned in the title but it’s impossible to trace it in the objective. Moreover, Non-inferiority clinical trials are designed to determine whether an intervention is not ‘unacceptably worse’ than a comparator (what’s the comparator?) by more than a prespecified difference, known as the non-inferiority margin (what’s the inferiority margin?). The selection of an appropriate margin is fundamental to non-inferiority trial validity.
- Please, state why a stepwise analysis was performed. A stepwise analysis is based on statistical criteria, and this is not always the best way to assess biological associations.
- In the discussion section, the two issues above should be considered.
Minor issues
- Please change through “P” or “p” for p
- Table 1. Add a footnote to explain what BDC and PTDC mean.
Author Response
D’Amuri A et al aimed to “assess the plasma levels of irisin at baseline and after aerobic exercise program comparing HIIT versus MICT, as well as described the specific determinants of irisin level among obese young healthy individuals.” The paper is interesting and well written and helps to disentangle the role of Irisin in biological systems.
REPLY: We thank the reviewer for the positive feedback.
However, there are a few issues that need attention.
Major issues
- A clear objective should be stated. A non-inferiority controlled randomized study is mentioned in the title but it’s impossible to trace it in the objective. Moreover, Non-inferiority clinical trials are designed to determine whether an intervention is not ‘unacceptably worse’ than a comparator (what’s the comparator?) by more than a prespecified difference, known as the non-inferiority margin (what’s the inferiority margin?). The selection of an appropriate margin is fundamental to non-inferiority trial validity.
REPLY: We thank the reviewer for the comment. We apologize for the misunderstanding due to the lack of a proper statement of the aim in the methods section. We have now clarified that this is a secondary analysis of the original non-inferiority interventional trial that was designed with a specific aim and already published (see ref. 25, D'Amuri A, et al. BMJ Open Sport Exerc Med. 2021;7:e001021). To avoid misleading the readers we have now clarified the aim in the Methods section.
…[For this secondary analysis aimed at studying the biological response of irisin to long-term aerobic exercise, we used data that come from a randomized, single-blind, single-centre, parallel-group, non-inferiority trial previously published that compares the efficacy of 12-weeks of HIIT and MICT in weight loss among obese (i.e., BMI between 30 and 55 kg/m2) healthy adults (≤50 years). The study design and the main findings concerning the primary objective of the original study has been already published [25].]
- Please, state why a stepwise analysis was performed. A stepwise analysis is based on statistical criteria, and this is not always the best way to assess biological associations.
REPLY: We thank the reviewer for the comment. As noted by the reviewer, a stepwise analysis based on statistical criteria is questioned as methodology for generating multivariate models when it comes to biological association. Surely, the stepwise approach is not ideal, but we tried to mitigate the intrinsic limitation of this approach as follows. We relied on the likely biological plausibility of the univariate association between covariates and dependent variable in the selection of the covariates to include in the multivariate model also accounting for the potential existence of collinearity. Then, the multivariable model was performed.
… [Based on biological plausibility, a set of covariates were included in a linear regression multivariate stepwise analysis to identify the factors independently associated with post training irisin level].
- In the discussion section, the two issues above should be considered.
REPLY: Please consider the above answers.
Minor issues
- Please change through “P” or “p” for p
REPLY: Amended.
- Table 1. Add a footnote to explain what BDC and PTDC mean.
REPLY: Added.